# The Moderating Effect of Flexible Work Option on Structural Empowerment and Generation Z Contextual Performance

**DOI:** 10.3390/bs13030266

**Published:** 2023-03-17

**Authors:** Daliah Taibah, Theresa C. F. Ho

**Affiliations:** 1Collage of Business Administration, University of Business and Technology, Jeddah 21361, Saudi Arabia; 2Azman Hashim Business School, Universiti Teknologi Malaysia, Kuala Lumpur 54100, Malaysia

**Keywords:** Generation Z, flexible work option, contextual work performance, structural empowerment

## Abstract

Leading and managing Generation Z can be a daunting task due to the distinct expectations, behaviours, and preferences they bring with them compared to prior generations. As such, when managing Gen Z workers, it is essential that leaders are aware of these variations to effectively manage their teams. Hence, this research endeavours to investigate the role of flexible work options on strengthening the relationship between structural empowerment and contextual work performance among Generation Z. This study employed a quantitative approach via an online questionnaire distributed to full- or part-time employed Gen Z workers in Jeddah, Saudi Arabia working within the wholesale and retail sectors. The purpose of this study is, therefore, to determine the impact of the moderating effect of flexible work options on the relationship between structural empowerment and Gen Z employee work performance. The findings suggest that H1, the moderating effect of the flexible work option on the opportunity and contextual work performance relationship, is not supported, while H2 and H3, the moderating effect of the Flexible work option on the relationships of support as well as information and contextual work performance, are both supported. However, it has a negative effect on the relationship between access to support and Gen Z contextual work performance and a positive effect on the relationship between access to information and Gen Z contextual work performance. The study’s findings highlight the necessary structural empowerment for enhancing Generation Z’s contextual work performance, offering useful information to management, policy makers, and the business as a whole.

## 1. Introduction

Employees that fall under the Generation Z category were born between 1995 and 2010 [1] and grew up during the digital revolution, a period of significant transformation in society. They are the generation that cannot exist without internet-connected devices, such as tablets, smartphones, and social media [2].

Lack of awareness of generational differences can hinder hiring and retention efforts, raise absenteeism, and negatively affect teamwork and leadership effectiveness. In addition, generational differences may cause conflict. Tensions and disputes may develop as a result of different views, behaviours, and cognitive processes, as well as a lack of understanding between parties [3]. Hence, leaders and managers need to be aware of the variations in workplace behaviours between generations in order to adjust their expertise and managerial practices to these changes [4].

Studies have indicated that Gen Z employees differ from members of Generation Y (Millennials) (born between 1980 and 1994) [5,6,7]. Generation Z workers need more structural empowerment from their leaders to perform better at work [8]. Hence, it is important that businesses discover ways to boost and improve Gen Z employees’ work performance. Furthermore, Generation Z is also more pragmatic, impatient, adaptable, braver, optimistic about the future, and has more realistic work expectations than Millennials because they do not understand the concept of struggling [5,6,7]. They constantly seek out new opportunities and motivations, and they have no fear of change [5,9]. Generation Z values openness, adaptability, independence, and individual liberty. Face-to-face communication is their preferred means of communication, and they insist on being educated, taken seriously, and having their ideas heard and acknowledged [6]. The climate at work needs to encourage mentoring, chances for career advancement, and an entrepreneurial culture. They also prefer places of employment with a social environment and flexible hours [6].

In order to adapt to the economic changes in the business environment of today, most Chief Executive Officers (CEOs) are looking to non-traditional work-life benefit initiatives. Schedule flexibility may have a good impact on employees’ health, work-life balance, and job happiness, according to studies [7,10,11]. As a result, employees would perform better on the job and have lower absence and attrition rates. Thus, the purpose of this study is to examine the impact of the moderating effect of the flexible work option on the relationship between structural empowerment and employee work performance.

### 1.1. Literature Review

A generation, often referred to as a generational cohort, is a group of individuals who had similar sociological and historical experiences at critical junctures in their development and were born within a specific time frame [3]. According to the generational theory, generational cohorts differ from one another due to the shared social circumstances and life experiences that allowed them to develop a common set of beliefs, standards, attitudes, and worldviews [12]. As a result, generational cohorts grow to be unique from one another. Professional literature uses slightly different categories and names to classify each generation as well as its time period [5]. The category employed in this study is largely based on the categorization used by [5], who define Generation Z as those born between 1995 and 2010.

There is a compelling need for a better understanding of the issues which influence Generation Z’s work performance as they are much different from previous generations [5,6], and they are more practical and more realistic than Millennials [5,6]. As a result, they embrace change and constantly seek out new challenges [5,9]; nonetheless, they lack work experience as well as problem-solving abilities and have not yet developed their decision-making abilities [6]. Therefore, numerous businesses criticize the quality of their performance [13].

Contextual performance refers to actions that “contribute to the production of a good or the provision of a service”, as opposed to task performance, which focuses on actions that are “needed to support the social fabric of the organisation” [14]. Additionally, it enhances team decision-making, ongoing professional development, employee impact, and solid community ties [15]. As a result, such performance boosts worker engagement and helps organizations achieve their objectives.

Another important factor that affects Generation Z’s work performance is employee empowerment [16]; empowered workers are more likely to achieve higher levels of productivity because they feel more in control of their work. Employee empowerment is, therefore, potentially a key factor that may bring about a number of positive effects, such as improved organizational responsiveness and increased employee satisfaction and productivity [17,18]. The workplace culture in Saudi Arabia is primarily perceived as authoritarian, with little room for autonomy [19]. Because of this, employee empowerment is only used by a small number of organizations and executives; however, this number is growing. Even yet, few studies on the impact of employee empowerment on performance have been carried out in Saudi Arabia [19,20,21,22], with the majority of these studies concentrating on the empowerment of [20,21,22]. While Reda et al. (2016) investigated the impacts of employee empowerment on performance, Ref. [19] investigated the impact of empowerment on workplace creativity.

Structural empowerment refers to “the actions taken by the leader to delegate the decision-making powers to subordinate” [18]. Because the barriers between managers and employees are reduced, there is better communication, information exchange, and employee involvement in decision-making [23,24]. The “theory of structural empowerment” was first put forth by Kanter in 1993. According to this notion, employee empowerment is “promoted in work environments that provide employees with access to information, resources, support, and the opportunity to learn and develop” [25]. Furthermore, structural empowerment emphasizes the power structures that enhance collaborative decision-making, continual professional growth, employee impact, and solid community ties. Consequently, it ensures that the organization’s goal, vision, and values are realized [15]. Therefore, structural empowerment seems to have a significant effect on employees’ performance [26].

This research focuses on Generation Z workers who have just entered the workforce and are, therefore, mostly regarded as junior or low-level staff. Particularly in the retail and wholesale sectors, these jobs require the least amount of paperwork and access to any financial resources. Therefore, it would be unnecessary for this study to investigate the impact of resource access on structural empowerment. Therefore, this study only focused on structural empowerment’s access in relation to opportunity, information, and support dimensions. Access to opportunity, as described by [27], is the capacity for professional development and advancement, as well as the chance to broaden one’s knowledge and skill set. While access to support is getting advice and comments from peers, superiors, and subordinates, as well as assistance with problem-solving [27,28]. Whereas access to information is having the formal and informal technical knowledge and competence required to carry out duties in the workplace successfully, as well as comprehending organizational decisions, policies, and the current situation [29].

Additionally, the effect of the flexible work option on performance has been studied a lot recently. However, to the best of this research’s knowledge, none of those studies have examined the moderating effect of the flexible work option on the relationship between structural employee empowerment and contextual work performance. Having the flexible work option improved work-life balance, quality of life, productivity, the ability to recruit and keep top talent, increased competitive advantage, and decreased employee turnover [30]. While flexitime gives employees greater control over their work, which boosts performance, productivity, and job satisfaction [30], more empirical research supports the benefits of telecommuting for raising output and enhancing employee happiness, work-life balance, and job satisfaction [10,11,30]. Because of this, many firms give workers more freedom and flexibility in order to increase productivity, find and keep talented workers, and boost competitiveness without sacrificing the organization’s interests [30,31].

Flexible scheduling helps empowerment’s effect on workers’ performance [32]. This study intends to investigate the moderating role of flexible work options in the link between structural employee empowerment and Generation Z contextual work performance due to the significance of flexible work options on employees’ performance. Since Gen Z workers will soon make up the majority of the workforce, it is important to consider ways to improve their work performance through empowerment and flexible work schedules based on the background outlined above.

However, the Arab culture has unique traits that rule managerial and leadership behaviour and, as a result, significantly influence the development of leadership traits [33]. This has been seen in numerous Saudi Arabian organizations where staying late is the norm. Many managers push their employees to put in more time and effort at work, which creates a work-life imbalance and increases the risk of burnout, both of which have a negative influence on Generation Z workers. As a result, employees’ physical and mental wellbeing would suffer, which would impair their relationships with their families. Numerous businesses have adopted flexible work arrangements to improve their competitiveness and recruit and retain outstanding employees [30,31]. Additionally, the flexible work option is a crucial concern for workers in the retail and wholesale sectors due to their unpredictable schedules and long working hours; these options would give workers in these sectors the chance to effectively respond to job demands and would motivate them to engage in their work [32].

### 1.2. Hypotheses Development

RefsStudies on the effects of employee empowerment, flexibility, and family-work balance on contextual job performance have found that giving employees flexible work options would increase their opportunity to have more time to polish their skills and increase their knowledge, which would benefit their work performance [32,34]. They also found that the retail and wholesale industries is characterized by a stressful working environment with busy work schedules and long working hours. This may lead to work-family imbalance, which causes unfavourable outcomes such as poor performance. Thus, it can be hypothesized that:

**H1.** *The flexible work option positively moderates the relationship between access to opportunity and Generation Z employees’ contextual performance*.

The flexible work option might provide employees with the impression that they may choose when and where to complete their tasks, further enhancing their contextual job performance. Giving employees with flexible work options would help them better manage the demands of both their home and professional lives [31]. Emotional support is a component of the access to support dimension, which could improve their contextual performance. Moreover, A study has investigated the use of flexitime as a management strategy and the level of satisfaction among its users [35]. Their investigation revealed that flexitime provided “autonomy to employees to harmonize work and non-work demands on their time, resulting in better workplace relations” [35]. Thus, flexible scheduling, or flexitime, increased the effect of access to support in enhancing employees’ contextual performance. Thus, it can be hypothesized that:

**H2.** *The flexible work option positively moderates the relationship between access to support and Generation Z employees’ contextual performance*.

Access to information encourages individuals to participate in organizational processes, which improves their contextual performance and results in higher-quality work. Moreover, the flexible work option can provide employees with the time they need to process information they have acquired and respond appropriately to it, which improves their contextual performance. organizational strategies, such as information sharing and offering flexible rotations and positions boosted workers’ creativity and productivity [36]. Therefore, it can be hypothesized that:

**H3.** *The flexible work option positively moderates the relationship between access to information and Generation Z employees’ contextual performance*.

## 2. Methodology

This research is an explanatory approach as it aims to explain how the flexible work option affects Generation Z’s performance. Explanatory research, such as this one, relies on the formulation of hypotheses that specify the type and direction of links between the variables under investigation [37]. Such studies’ data are usually quantitative, which requires statistical tests to determine the relationship’s validity [37]. Thus, quantitative data were collected for this study, while the questions forming the survey will be based on previous research.

### 2.1. Variables and Measures

A preliminary questionnaire was adopted from previous studies for this research and consisted of five sections with a total of 33 items to examine the relationship between the independent variable, structural empowerment (adopted from [28]), the dependent variable, employee contextual performance (adopted from [38]), and the moderator, the flexible work option (from [31]) (Table 1). The first section of the questionnaire is an explanation of the questionnaire and the ethical obligation of the researcher. The second section includes participants’ demographic items. Since the target sample is the Generation Z workforce, this section is very important to this research. It consists of six items related to demographics; age, gender, education, work type, work experience, and if they are currently in their first job. This segment was self-developed. All items in Sections three, four and five are measured on a five-point Likert scale where one is ‘strongly disagree’ and five is ‘strongly agree’. Overall, the independent, dependent, and moderating variables in this study are:  i.Dependent variable: Generation Z’s Employee Contextual Performance ii.Independent variable: Structural Empowermentiii.Moderating variable: Flexible Work Option

### 2.2. Sample and Data Collection

An online survey was used in this study to gather information from Generation Z workers in Jeddah, Saudi Arabia’s retail and wholesale sectors. A total of 165 people answered the distributed online survey. However, just 109 met the sample’s requirements. This study also used nonprobability sampling. More precisely, this study used convenience sampling since it is not feasible to include every subject in the study because the population is nearly finite. Considering that the questionnaire was online, a provision was made that forbade respondents from submitting their responses until all questions had been addressed. There are no missing data in this study as a result.

Findings also show that around eight percent (n = 9; percentage = 8.3%) of respondents were between the ages of 15 to 18, while the majority of respondents were almost split between the ages of 19 to 23 (n = 54; percentage = 49.5%), and the ages of 24 to 27 (n = 46; percentage = 42.2%). Moreover, there was an almost similar participation from both genders, male and female, with the males’ participation being slightly higher (male n = 61; percentage = 56%, females n = 48; percentage = 44%). Education varied among participants, with the highest percentage being those with bachelor’s degrees (n = 71; percentage = 65.1%) and most respondents having full-time jobs (n = 74; percentage = 67.9%). Furthermore, more than 50 percent of participants are occupying their first jobs and do not have previous work experience (n = 64; percentage = 58.7).

### 2.3. Data Analysis Methods and Techniques

This study used multiple linear regression data analysis. Moreover, composite reliability was used to measure the internal consistency reliability of the questionnaire while collected data from the questionnaire were analysed systematically using Smart PLS 3.2.1 and IBM SPSS Statistics version 21.0. Data analysis in this research was carried out using structural equation modelling (SEM). SEM is a group of statistical methods which “enable researchers to incorporate unobservable variables measured indirectly by indicator variables. They also facilitate accounting for measurement error in observed variables” [39].

### 2.4. Moderator Analysis

The moderator is considered an independent variable, which changes the strength or the direction of a relationship between two variables in the model [40]. While the mediator is a variable that is “in a causal sequence between two variables” [41], the moderator, however, is not part of it.

There are several approaches for moderation analysis when using PLS–SEM. moderator analysis using the two-stage approach PLS–SEM method will be performed in this study to test the significance of its moderator via SmartPLS 3, GmbH, Germany software. This approach will be used because it is normally the most adaptable approach and is usually the most preferred.

## 3. Analysis of Findings

### 3.1. Demographic Profile

Table 2 demonstrates the demographic numbers of the responses and their percentages. The findings show that even though 165 individuals responded to the questionnaire, only 109 fit the sample criteria. The minimum sample size required for this study was 103, which exceeds the requirement. Hence, this study proceeds with 109 respondents.

Findings also show that around eight percent (n = 9; percentage = 8.3%) of respondents were between the ages of 15 to 18, while the majority of respondents were almost split between the ages of 19 to 23 (n = 54; percentage = 49.5%), and the ages of 24 to 27 (n = 46; percentage = 42.2%). Moreover, there was an almost similar participation from both genders, male and female, with the males’ participation being slightly higher (male n = 61; percentage = 56%, females n = 48; percentage = 44%). Education varied among participants with the highest percentage being those with bachelor’s degrees (n = 71; percentage = 65.1%) and most respondents having full-time jobs (n = 74; percentage = 67.9%). Furthermore, more than 50 percent of participants are occupying their first jobs and do not have previous work experience (n = 64; percentage = 58.7).

### 3.2. Assessment of Measurement Model

The measurement model for this study includes the assessment of indicator reliability, internal consistency reliability, convergent validity, and discriminant validity, as suggested by [42,43]. Indicator reliability is measured by the “Indicator loadings”. The cut-off values of indicator loadings are usually below 0.5 [43,44]. Table 3 shows the indicator loadings for retained items. The lowest loading value is 0.575; therefore, none of the items need to be removed.

As for the internal consistency reliability, it is measured in this research by Cronbach’s alpha and the consistent reliability coefficient (rho_A). Higher values above 0.7 display higher levels of reliability [43]. Table 4 shows the findings of this research, where the lowest value is 0.805 for Cronbach’s alpha and 0.813 for rho_A. Therefore, the values are satisfactory. The convergent validity is measured by using “the average variance extracted” (AVE) [40]. The acceptable values of AVE are 0.50 or higher [42,43]. Table 4 shows that all constructs have AVE above 0.5 after the removal of items CON1, CON6, CON9 and FWO1 that have a low-value AVE (Table 3). This shows that the convergent validity for this research is significant. Another measure of convergent validity is composite reliability (CR) [40]. CR values should be 0.7 or higher [43]. The findings of this study, shown in Table 4, are all above 0.7. Thus, the convergent is significant.

Lastly, discriminant validity was verified using the SMART–PLS heterotrait-monotrait (HTMT). If the values of HTMT are close to 1, this indicates that there is a lack of discriminant validity [40]. Some scholars, such as [43,45], suggested that the threshold of value should not be above 0.90. Table 5 shows the findings of the HTMT of this study. All the values are within the suggested threshold, hence indicating discriminant validity. In conclusion, this study is discriminately valid, and the measurement of its constructs is reliable and valid.

### 3.3. Moderator Testing

Data were acquired by running the SEM–PLS algorithm, whereas the significant *p*-value and the t-value were obtained through bootstrapping using 5000 subsamples. Table 6 shows the findings of the hypothesis testing. The results of the path coefficients in all relationships were more than zero (between −1.00 and +1.00). Two of the three hypotheses showed a positive and significant relationship with *p*-values of less than 0.05 and t-values of more than 1.645: H2 and H3.

There are several approaches for moderation analysis when using PLS–SEM; however, the two-stage approach is normally the most adaptable approach and is usually the most preferred. This can be implemented using SmartPLS 3 software [40]. This study proposes that the flexible work option moderates the relationship between the three dimensions of structural empowerment and contextual work performance. Table 6 illustrates the findings of this study’s moderator testing. The findings suggest that H1, the moderating effect of the flexible work option on the opportunity and contextual work performance relationship, is not supported (t = 0.377; *p* = 0.353), while H2 and H3, the moderating effect of the flexible work option on the relationships of support, as well as information and contextual work performance, are both supported (t = 1.705; *p* = 0.44 and t = 1.905; *p* = 0.028, respectively; Table 3).

#### 3.3.1. H2: The Flexible Work Option Moderates the Relationship between Access to Support, and Generation Z’s Employee Contextual Performance

The H2 beta coefficient is negative (β = −0.173). This indicates that the relationship between support and contextual work performance is stronger when the flexible work option is lower. The effect size is 0.028, which indicates a small effect size.

The graph presented in Figure 1 indicates the impact of the flexible work option on the relationship between support and contextual work performance. The higher the flexible work option, the weaker the relationship between support and contextual work performance. Therefore, the flexible work option negatively moderates this relationship.

#### 3.3.2. H3: The Flexible Work Option Moderates the Relationship between Access to Information, and Generation Z’s Employee Contextual Performance

On the other hand, the H3 beta coefficient is positive (β = 0.272). This indicates that the relationship between information and contextual work performance is stronger when the flexible work option is higher. The effect size is 0.061, which also indicates a small effect size but is relatively higher than in H2.

The graph presented in Figure 2 indicates the impact of the flexible work option on the relationship between information and contextual work performance. The higher the flexible work option, the stronger the relationship between information and contextual work performance. Therefore, the flexible work option positively moderates this relationship.

In summary, the flexible work option has no moderating effect on the relationship between access to opportunity and Gen Z’s contextual work performance. (H1) has a negative moderating effect on the relationship between access to support and Gen Z’s contextual work performance (H2), and a positive effect on the relationship between access to information and Gen Z’s contextual work performance (H3).

## 4. Discussion of Findings

### 4.1. Would Flexible Work Options Moderate the Relationship between Structural Empowerment and Generation Z Employees’ Contextual Work Performance?

The findings of this research have shown that flexible work options moderate the relationship between two out of three of the structural empowerment dimensions and Generation Z employees’ contextual work performance.

### 4.2. H1: Flexible Work Option Moderates the Relationship between Access to Opportunity, and Generation Z Employee Contextual Performance

The findings have shown that H1 is not significant. Therefore, the flexible work option does not moderate the relationship between access to opportunity structural empowerment, and Generation Z’s employee contextual performance. These findings are inconsistent with previous studies conducted by [32,34], who have found that the retail and wholesale industries are characterized by a stressful working environment with busy work schedules and long working hours. This may lead to work-family imbalance, which causes unfavourable outcomes, such as poor performance. In addition, Generation Z workforce prefers a working environment which allows for flexible schedules and work-life balance, which, consequently, increases their contextual job performance [31]. However, the findings of this research show otherwise; the flexible work option’s moderating effect on this relationship is not significant. Generation Z members look for jobs in organizations that “offer not only better compensation and benefits, but also have positive sustainability policies and activities, and take into consideration their well-being” [46]. This generation looks for working environments that provide them with a work-life balance [6]. Thus, if a Generation Z member was structurally empowered by an access to opportunity, the flexible work option would have no effect on them.

### 4.3. H2: Flexible Work Option Moderates the Relationship between Access to Support, and Generation Z Employee Contextual Performance

The findings have shown that H2 is not supported and that the flexible work option negatively moderates the relationship between access to support structural empowerment and Generation Z’s employee contextual performance. This indicates that the relationship between support and contextual work performance is stronger when the flexible work option is lower. This is inconsistent with the previous studies of that found that providing flexible work options would allow employees to better handle the competing stresses between home and work demands [31,35]. Furthermore, it gave “autonomy to employees to harmonize work and non-work demands on their time, resulting in better workplace relations”. However, Generation Z prefers face-to-face communication, mentoring, and having their responses heard [6]. Thus, Generation Z members may perceive the flexible work option as a hindrance to getting access to support. Hence, flexible scheduling, or flexitime, decreases the effect of access to support in enhancing employees’ contextual performance.

### 4.4. H3: Flexible Work Option Moderates the Relationship between Access to Information Structural Empowerment and Generation Z Employee Contextual Performance

The findings have shown that H18 is significant and that the flexible work option positively moderates the relationship between access to information structural empowerment and Generation Z’s employee contextual performance. This indicates that the relationship between information and contextual work performance is stronger when the flexible work option is higher. This is consistent with the previous studies of that found that organizational practices, such as providing information, providing a positive work environment and giving flexible rotations and roles, increased employees’ innovation and performance [32,36]. Therefore, leadership provides a positive work environment, which encourages employees to perform contextually better.

## 5. Implications and Contributions

The empirical findings of this current study have a notable contribution to future leadership research, Employee Structural Empowerment, Flexible work option, and Generation Z employees’ performance. The following points summarises the Implications and Contributions of the current study:

### 5.1. Implications and Contributions to Theory

This research contributes to the literature on the moderating effects of the flexible work option on Generation Z’s contextual work performance, specifically, for employees in the retail and wholesale industries, as it may be considered an important issue for them due to their irregular schedules and long working hours [30,31].

Furthermore, it has been hypothesized that Generation Z seeks to have a balance between work and family more than the previous generations [16]. However, until the time of this research, there have not been any published studies that focus on the moderating effect of the flexible work option on the relationship between structural employee empowerment and the contextual work performance of a specific generation.

Furthermore, the findings of this study showed that the flexible work option has a moderating effect on two of the three relationships between the structural empowerment dimensions and Generation Z’s contextual work performance. It was found that the flexible work option positively moderates the relationship between access to information structural empowerment and Generation Z’s employee contextual performance. In contrast, it negatively moderates the relationship between access to support structural empowerment and Generation Z’s employee contextual performance. However, it does not moderate the relationship between access to opportunity structural empowerment and Generation Z’s employee contextual performance. This indicates that the higher the flexible work option is, the stronger the relationship between access to information and Generation Z’s contextual work performance would be. Oppositely, the lower the flexible work option is, the stronger the relationship between access to support and Generation Z Contextual work performance would be. Therefore, leaders must weigh the amount of flexible work options they would provide to Generation Z employees.

### 5.2. Implications and Contributions to Practice

From the practical perspective, the findings of this study provide useful information to management, policy makers and organizations as a whole by highlighting the appropriate structural empowerment needed to improve Generation Z’s contextual work performance. In addition, the findings of this study can help organizations to carefully weigh the costs and benefits related to creating policies and help employees from all generations, including Generation Z, to reach their highest levels of capability.

### 5.3. Limitations and Suggestions for Future Research

The design of this study, like most studies, has limitations, which present possibilities for further investigation. First, the scope of this study was limited to Generation Z working in the wholesale and retail industries in Jeddah city. Since different industries or different cities may give different results, generalization and extending the findings of this study to other industries or cities in Saudi Arabia is not possible. Hence, the replication of the study on different cities and industries would produce a more generalized picture. In addition, further research should examine the validity of this study’s findings by replicating and testing respondents from other generations as well as other countries. Second, this study’s methodology was of a quantitative nature. Therefore, it is recommended that a future qualitative study should be conducted to extend upon the perspective of this study and explore other dimensions that influence Gen Z’s performance. Finally, future studies may consider how the flexible work option affects different personality types of Gen Z employees and its impact on their performance.

## 6. Conclusions

This study examined the moderating effect of the flexible work option on the relationship between structural empowerment and Generation Z’s contextual work performance. Generation Z employees are different from previous generations. Thus, leaders need to be aware of such differences when managing the Generation Z workforce. The empirical findings of this current study have showed that the flexible work option has a moderating effect on two of the three relationships between structural empowerment dimensions and Generation Z’s contextual work performance; the flexible work option positively moderates the relationship between access to information structural empowerment and Generation Z’s employee contextual performance. In contrast, it negatively moderates the relationship between access to support structural empowerment and Generation Z’s employee contextual performance. However, it does not moderate the relationship between access to opportunity structural empowerment and Generation Z’s employee contextual performance. Such results have a notable contribution to future leadership research, employee structural empowerment, the flexible work option, and Generation Z’s employees’ performance. In addition, the findings of this study provide useful information to management, policy makers and organizations as a whole by highlighting the appropriate leadership characteristics and structural empowerment needed to improve Generation Z’s contextual work performance.

Since this research was limited to Generation Z working in the wholesale and retail industries in Jeddah city, it is recommended that further future research is needed in other Saudi cities and industries, which would produce a more generalized picture and generate a better understanding of the effect of the flexible work option on Generation Z’s performance. Moreover, it is recommended that future qualitative research should be conducted to extend upon the perspective of this study. Finally, leaders must carefully weigh the costs and benefits related to creating policies and help employees from all generations, including Generation Z, to reach their highest levels of capability.

## Figures and Tables

**Figure 1 behavsci-13-00266-f001:**
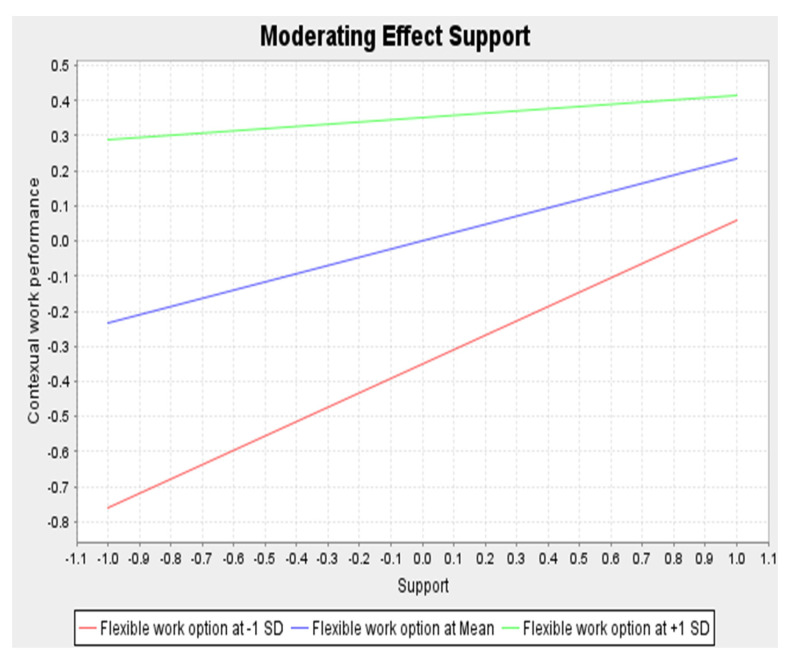
Impact of flexible work option on the relationship between support and contextual work performance.

**Figure 2 behavsci-13-00266-f002:**
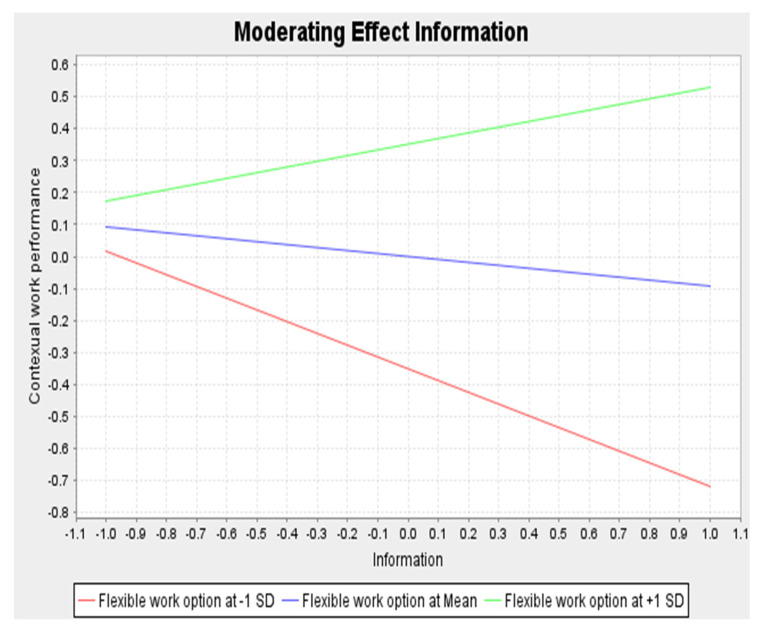
Impact of flexible work option on the relationship between information and contextual work performance.

**Table 1 behavsci-13-00266-t001:** Summary of key constructs, sources of questions, and the number of items.

Section	Variable	Dimension	No of Items	Source
2	Demographic		6	Self-Developed
3	employees’ contextual work performance		12	[28]
4	Structural Employee Empowerment	Access to opportunity	3	[38]
		Access to support	3	
		Access to information	3	
5	flexible work option		6	[31]

**Table 2 behavsci-13-00266-t002:** Respondents’ Demographic.

Demographics	Frequency (n)	Percentage
**Age**		
15–18	9	8.3%
19–23	54	49.5%
24–27	46	42.2%
**Gender**		
Male	61	56%
Female	48	44%
**Education**		
High school	31	28.4%
Diploma	2	1.9%
Bachelor’s degree	71	65.1%
Other	5	4.6%
**Work type**		
Full time	74	67.9%
Part time	35	32.1%
**Work experience**		
0–2 years	73	67%
3–5 years	30	27.5%
more than 5 years	6	5.5%
**Is this your first job?**		
Yes	64	58.7%
No	45	41.3%

**Table 3 behavsci-13-00266-t003:** Indicator loadings values.

Construct	Number of Items	Item Deleted	Loadings for Retained Items
Contextual work performance	9	CON1, CON6, CON9	CON2 0.697
			CON3 0.686
			CON4 0.666
			CON5 0.759
			CON7 0.707
			CON8 0.693
			CON10 0.747
			CON11 0.742
			CON12 0.673
Opportunity	3	-	SEE1 0.851
			SEE2 0.915
			SEE3 0.866
Support	3	-	SEE4 0.887
			SEE5 0.934
			SEE6 0.842
Information	3	-	SEE7 0.894
			SEE8 0.904
			SEE9 0.901
Flexible work option	5	FWO1	FWO2 0.642
			FWO3 0.629
			FWO4 0.813
			FWO5 0.839
			FWO6 0.821

**Table 4 behavsci-13-00266-t004:** Construct Reliability and Validity.

Construct	Cronbach’s Alpha	Rho_A	Composite Reliability	Average Variance Extracted (AVE)
Information	0.883	0.890	0.927	0.809
Opportunity	0.851	0.854	0.910	0.771
Support	0.866	0.872	0.918	0.789
Contextual work performance	0.876	0.883	0.900	0.502
Flexible work option	0.805	0.813	0.867	0.569

**Table 5 behavsci-13-00266-t005:** Discriminant Validity-HTMT.

		1	2	3	4	5
1.	Contextual work performance					
2.	Flexible work option	0.716				
3.	Information	0.574	0.729			
4.	Opportunity	0.710	0.701	0.819		
5.	Support	0.677	0.675	0.770	0.770	

**Table 6 behavsci-13-00266-t006:** Findings of moderator testing.

Hypothesis	Relationship	Std Beta	SD	t-Value	*p* Values	CI		Decision	F^2^	F^2^ Results
H1	Moderating Effect on Opportunity -> Contextual work performance	−0.034	0.09	0.377	0.353	−0.164	0.125	Not Supported	0.002	No effect
H2	Moderating Effect on Support -> Contextual work performance	−0.173	0.101	1.705 **	0.044	−0.340	−0.015	Supported	0.028	Small
H3	Moderating Effect on Information -> Contextual work performance	0.272	0.143	1.905 **	0.028	0.005	0.457	Supported	0.061	Small

** *p* < 0.05.

## Data Availability

The data presented in this study are available on request from the corresponding author. The data are not publicly available due to the fact that it is a part of a PhD Dessertation.

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
