# Peer review of "The Moderating Effect of Flexible Work Option on Structural Empowerment and Generation Z Contextual Performance"

_behavsci, 2023, doi:10.3390/bs13030266_

Round 1

Reviewer 1 Report

Minor corrections, see examined manuscript.

Author Response

Dear Reviewer,

Thank you for your kind review of my article.

I have made the changes you have suggested; mainly, I have added summery of findings and reviesed the abstract, and revised the third paragraph in the introduction and fixed the typos in variouse sentences and other grammar mistakes which you have kindly highlighted. 

Once again, Thank you

Reviewer 2 Report

Review: the moderating effect of flexible work option on structural employment and generation said contextual performance

This paper focuses upon the differences in leading and managing Gen Z cohort and especially the utility of flexible work in encouraging empowerment and subsequent performance in workers in the wholesale/retail industry in Jeddah, Saudia Arabia.

Introduction

No doubt societal transformations caused by digitisation et cetera have significantly impacted the attitudes and behaviours of Gen Z workers and leaders need to adapt accordingly. Appropriate literature is cited here, including a greater need for structural empowerment. Various traits have been proposed as differentiating Gen Z, including a preference for face-to-face communication and the need for validation of their opinions.

Literature review

Again I find appropriate supporting literature here. Page 2 line 74 repeats the word practical. You mention Millennials: have these been clearly defined?

I like the logic in linking Saudia Arabia work culture to empowerment and Gen Z aspirations. Structural empowerment here means access to opportunity, support, and information.

Page 3 line 133: “in addition,…. Contextual work performance.” These are clumsy sentences and need rewriting.

There are several statements in the 1st paragraph that need references.

The hypotheses seem reasonable

Method

 You probably need a bit more justification for using a quantitative approach.

The questionnaire was sent to 165 workers online and there are 33 items adopted from several authors, as presented in table 1. It would be good to see a copy of the questionnaire in the appendix. 109 relevant questionnaires were received. The participants are mostly young, equally split between male and female, undergraduate qualified and full-time workers. Probably the information which is still to be presented in 3.1 and table 2 would be sufficient.

How were the respondents targeted? What was the sampling method? Information must be provided on this

Analysis/findings

Page 8 line 29:” convergent”?

Regarding the hypotheses, that flexible work option will moderate access to opportunity and performance is not supported. The other 2 hypotheses are supported, although there is a negative relationship in terms of hypotheses two.

Figures 1 and 2 are helpful

it might be a good idea to clearly state all of the 3 findings in a succinct fashion at the end of this section

discussion/findings; you have called discussion and findings but I think you mean discussion of findings

The results seem to be discussed appropriately. On page 11 I did get a bit confused about this negative relationship between flexible work option and access to support and performance but I can follow your logic here .

implications et cetera

suddenly you introduce the concept of corporate survival: has been mentioned before? If not, you will need to talk about it earlier. Otherwise fine.

conclusion

 seems okay to me

 in general

Presentation is lacking. There seem to be different fonts in parts of the introduction, for example. This paper needs careful proofreading. Regarding the references: the 5th reference has the last names and 1st names confused. There are inconsistencies. The reference to Thompson is repeated. Also, the in text referencing on page 3 at line 1112 to Larkin looks strange

There are numerous typos throughout

Author Response

Dear Reviewer,

Thank you for youe kind review. Below are my responces to your review:

Point 1: Page 2 line 74 repeats the word practical.

Responce 1: noted and fixed

Point 2: You mention Millennials: have these been clearly defined?

Responce 2: Diffenition has been added

Point 3: Page 3 line 133: “in addition,…. Contextual work performance.” These are clumsy sentences and need rewriting.

Responce 3: noted and fixed

Point 4: There are several statements in the 1st paragraph that need references.

Responce 4: noted and reference has been added

Point 5:  The questionnaire was sent to 165 workers online and there are 33 items adopted from several authors, as presented in table 1. It would be good to see a copy of the questionnaire in the appendix. 

Responce 5: I need to check how to attach the appendix to the article with the journal

Point 6: How were the respondents targeted? What was the sampling method? Information must be provided on this

Responce 6: noted and fixed

Point 7: Page 8 line 29:” convergent”?

Responce 7: I did not understand the point. 

Point 8: it might be a good idea to clearly state all of the 3 findings in a succinct fashion at the end of this section

Responce 8:  noted and fixed

Point 9: discussion/findings; you have called discussion and findings but I think you mean discussion of findings

Responce 9:  noted and fixed

Piont 10: suddenly you introduce the concept of corporate survival: has been mentioned before? If not, you will need to talk about it earlier. Otherwise fine.

Responce 10:  the concept of corporate survival has been deleted

Point 11: Presentation is lacking. There seem to be different fonts in parts of the introduction, for example. This paper needs careful proofreading.

Responce 11: proofreadeing has been done and font was checked.

point 12: Regarding the references: the 5th reference has the last names and 1st names confused. There are inconsistencies. The reference to Thompson is repeated. Also, the in text referencing on page 3 at line 1112 to Larkin looks strange. There are numerous typos throughout

Responce 12: references were checked and fixed. 

Thank you for your time and efforts

Round 2

Reviewer 2 Report

2nd review: the moderating effect of flexible work option on structural empowerment and generation Z contextual performance

I can see that the abstract has been boosted.

Introduction

There is significant additional information in the introduction including information about generation Y as requested. on page 2, I’m not quite sure about the statement regarding generation Z that they insist upon being educated. Perhaps a reference would help here.

Literature review

 the numbering seems inconsistent from page 2 to page 3. I still assert that the sentences at the bottom of page 3 commencing in addition up until contextual work performance need to be written more substantially. The hypotheses have additional information which is good.

Method

there still needs to be further justification about the use of a quantitative approach here. Why not choose qualitative in order to explore the motivations behind the attitudes of the Gen Z cohort?

Some additional information is provided about the sample, but was it really disseminated to all Internet users in retail and wholesale sector in Jeddah? I would be surprised if that were the case.

Findings

nice to see a summary section on page 10. I feel as though “Flexible work option” needs to be preceded by “a” throughout the manuscript; it just feels odd without a word before it.

Discussion

I still say it should be discussion of findings not discussion and findings because you have already had analysis of findings. Pleased to see that corporate work survival has been removed.

References

please check the 5th reference, which is still incorrect

Author Response

Thank you for your kind review.

Point 1: on page 2, I’m not quite sure about the statement regarding generation Z that they insist upon being educated. Perhaps a reference would help here.

Response 1: reference has been added

Point 2: the numbering seems inconsistent from page 2 to page 3. I still assert that the sentences at the bottom of page 3 commencing in addition up until contextual work performance need to be written more substantially.

Response 2: the sentence has been rephrased, citation style has been changed upon the request of the journal.

Point 3: there still needs to be further justification about the use of a quantitative approach here. Why not choose qualitative in order to explore the motivations behind the attitudes of the Gen Z cohort?

Response 3: done

Point 4: Some additional information is provided about the sample, but was it really disseminated to all Internet users in retail and wholesale sector in Jeddah? I would be surprised if that were the case.

Response 4: the statement was changed

Point 5: I feel as though “Flexible work option” needs to be preceded by “a” throughout the manuscript; it just feels odd without a word before it.

Response 5: it is written without “a” throughout all literature

Point 6: I still say it should be discussion of findings not discussion and findings because you have already had analysis of findings.

Response 6: done

Point 7: References please check the 5th reference, which is still incorrect

Response 7: noted and fixed